# Experimental Research and Numerical Analysis of CFRP Retrofitted Masonry Triplets under Shear Loading

**DOI:** 10.3390/polym14183707

**Published:** 2022-09-06

**Authors:** Houria Hernoune, Benchaa Benabed, Rajab Abousnina, Abdalrahman Alajmi, Abdullah M GH Alfadhili, Abdullah Shalwan

**Affiliations:** 1Department of Civil Engineering, University of Yahia Fares, Medea 26000, Algeria; 2Civil Engineering research Laboratory, Department of Civil Engineering, University of Amar Telidji, Laghouat 03000, Algeria; 3School of Engineering, Faculty of Science and Engineering, Macquarie University, 2109 Sydney, Australia; 4Department of Power and Desalination Plants, Ministry of Electricity and Water, Kuwait City 12010, Kuwait; 5Department of Manufacturing Engineering Technology, Public Authority for Applied Education and Training, Kuwait City 13092, Kuwait

**Keywords:** masonry triplets prisms, concrete damaged plasticity (CDP), CFRP strips, detailed micro-modelling approach (DMM), CFRP-reinforced masonry specimens, extended finite element method (XFEM)

## Abstract

This paper presents an experimental and numerical study into the shear response of brick masonry triplet prisms under different levels of precompression, as well as samples reinforced with carbon fiber-reinforced polymer (CFRP) strips. Masonry triplets were constructed with two different mortar mix ratios (1:1:3 and 1:1:5). In this study, finite element models for the analysis of shear triplets are developed using detailed micro-modelling (DMM) approach and validated with the experimental data. The failure mechanisms observed in the masonry triplets were simulated using a coupled XFEM-cohesive behaviour approach in ABAQUS finite element software. The nonlinear behaviour of mortar and brick was simulated using the concrete damaged plasticity (CDP) constitutive laws. The cohesive element with zero thicknesses was employed to simulate the behaviour of the unit–mortar interfaces. The extended finite element method (XFEM) was employed to simulate the crack propagation in the mortar layer without an initial definition of crack location. CFRP strips were simulated by 3D shell elements and connected to masonry elements by an interface model. The changes in failure mechanism and shear strength are calculated for varying types of mortar and fiber orientation of CFRP composite. Based on this study, it was concluded that the ultimate shear strength of masonry triplets is increased due to the external bonding of CFRP strips. The performance of masonry specimens strengthened with CFRP strips is assessed in terms of gain in shear strength and post-peak behaviour for all configurations and types of mortar considered. The comparison of FE and experimental results proved that the models have the potential to be used in practice to accurately predict the shear strength and reflect damage progression in unreinforced and CFRP-reinforced masonry triplets under in-plane loading, including the debonding of the CFRP reinforcement. Additionally, XFEM was found to be a powerful technique to be used for the location of crack initiation and crack propagation in the mortar layer.

## 1. Introduction

Masonry structures represent a large part of existing structures around the world, most of which are located in regions of high seismic events. Various seismic events have emphasised the vulnerability of this kind of building, causing a gradual increase in researchers’ development and improvement of retrofitting techniques. The lateral load resistance of masonry structures is principally due to the in-plane shear strength of the masonry elements. Hence, a detailed analysis of the in-plane shear behaviour of masonry thus becomes necessary. The earthquake performance of a masonry wall is very sensitive to the properties of its constituents, namely masonry units, and mortar. The shearing strength of masonry mostly depends on the bond at the contact surface between the masonry unit and the mortar. Thus, it is very important to improve the shear behaviour of masonry buildings. Mortar joints usually represent the weakest part of masonry and their non-linear behaviour in tension and shear can heavily affect the overall response of the composite.

The prediction of the structural capacity of unreinforced masonry buildings under combined compression and shear loading requires an accurate characterisation of the masonry strength and general response under shear stresses. The experimental investigations of shear strength parameters typically relies on shear walls, or on standard triplets [1]. According to standards BS EN [2], the triplet shear test allows the determination of the in-plane shear strength of the horizontal bed joint in masonry. The triplet test has been largely applied in both research and industrial fields and is used to characterise many different materials [3,4]. The allowable pre-compression level has already been investigated by various researchers; experimental evidence has pointed out that this parameter is another important factor that influences the masonry elements’ in-plane shear strength and deformability [5]. Similarly, the same results were obtained by Capozucca [6]. Through these studies, it appeared that normal compressive stresses acting on the interface affect both the peak shear stress and residual strength.

In order to extend the life of such structures, strengthening or repairing by implementing new techniques have been developed. Many of these strengthening techniques, including the use of fiber-reinforced polymer and (FRP) composites, have aided in reinforcing masonry structures. These composites are manufactured in different features depending on the fiber material type, such as carbon fiber (CFRP), glass (GFRP), and aramid (AFRP). FRP strips offer the possibility of application by gluing on the outside surface EB (externally bonded) or inserting inside the groove of element using the near-surface mounted (NSM) technique. The use of FRP composites embedded into polymetric matrix such as polyester, epoxy, and mortar mix, can be an effective solution for retrofitting the structure walls in seismic areas.

Carbon fiber-reinforced polymers (CFRP) and glass fiber-reinforced polymers (GFRP) have been widely used in many applications on different kinds of masonry [7,8,9]. Son, Seung-Hwan et al. [10] investigated the enhancement of the in-plane strength and ductility of UMWs using GFRPU. They illustrated that the GFRPU reinforcement of the masonry wall leads to enhanced load-carrying capacity, ductility, and energy absorption. El-Diversity et al. [11] conducted an experiment to verify the reinforcement effect of masonry walls, reinforced by ferrocement and glass fiber-reinforced polymer (GFRP) complex systems, on the in-plane recurrent load. It was noticed that the reinforcement technique increased the ductility and energy absorption capacity by 33 to 85% and the lateral resistance performance by 25 to 32%.

In recent decades the rehabilitation and retrofitting of masonry walls using externally bonded carbon fiber-reinforced polymer (CFRP) strips is considered to be the most successful and economical technique available [9,12,13]. The use of CFRP composites to strengthen and rehabilitate masonry has been examined by many authors. Their experimental studies have evaluated the effect of the retrofitting configuration and the types of CFRP composites and masonry components for small scale and full-scale masonry walls [14,15,16,17,18]. These experiments proved how efficiently the CFRP retrofitting system can improve the shear strength of unreinforced masonry walls [19,20]. All these studies have shown that the CFRP strengthening technique will increase the seismic capacity, stiffness, and ductility of masonry walls subjected to in-plane lateral loading or out-of-plane loading [9,21,22,23,24]. Hernoune et al. [9] conducted a series of experimental tests on URM walls, which illustrated the effectiveness of the CFRP retrofitting system in improving the shear strength of unreinforced masonry with a factor of 3 to 4. The effect of different variables including the type of masonry, CFRP, and retrofitting configuration on the performance of retrofitted masonry can be studied in an economical way (to reduce the time, cost, and effort) through finite element analysis [9].

Debonding is the most common failure and one of the undesirable failure mechanisms in FRP-strengthened masonry structures; if this rupture occurs, the total strength of the reinforcement cannot be exploited. Generally, this failure mostly occurs when the bonding between the reinforcement composite and masonry interface is lost. Most bonding studies found in the literature relate to the bond behaviour of FRP to concrete, with only a few researchers quite recently noticing the FRP-masonry bond behaviour [25,26,27], which involves the analysis of various parameters, such as mechanical properties of natural stone or bricks, composite systems, bond strength, bonded length, the type of the test, and the FRP width [26]. The same applies for the numerical analysis of this problem and only a limited number of studies can be found on this particular subject. In masonry panels and walls under in-plane shear loading, debonding initially starts at an intermediate location of the FRP strips, where the main diagonal crack developed crosses the FRP reinforcement. Later, the final brittle failure occurs by debonding of the overlays from the end of the strips [28]. All the research reported in the literature has confirmed that externally bonded FRP strips can increase the strength and deformation capacity of masonry walls subjected to both out-of-plane and in-plane loading. Furthermore, the debonding of the FRP composites from masonry substrates is one of the main failure modes.

In this type of research, several authors confirmed that the combination of experimental studies and numerical approaches is required so as to expand and enhance the current studies and to gain a more thorough understanding about the behaviour of brick masonry structures [29]. According to the literature, the modelling of masonry structure behaviour can be done through two different numerical approaches: macro-modelling and micro-modelling. Regardless of this, micro-modelling is among the most precise and reliable tools for modelling the behaviour of masonry walls subjected to various stress conditions, particularly in terms of their local response. The use of micro-modelling in numerical simulations of masonry wall in-plane and out-of-plane behaviour has produced satisfactory results [30]. The authors used a detailed micro-model based on a cohesive constitutive model for interface elements, and a damage plasticity model for mortar joints and units to reproduce the behaviour observed in diagonal compression tests of brick masonry assemblages. Several authors have recently used the same micro-model in their studies to numerically reproduce the shear response of brick masonry walls [31]. The masonry structural response simulation must define numerical strategies that accurately reproduce the brick rupture and the degradation experienced by the brick–mortar shear response. The implementation and evolution of these models, on the other hand, represent the most natural and realistic way to simulate masonry structures, allowing for the accurate reproduction of structural response based on the actual mechanical properties of elements, materials, and interfaces.

The use of a detailed micro–model for predicting masonry structural responses accentuates a detailed experimental description of the material constituents (mortar and bricks) as well as the unit–mortar interfaces [32]. The approach chosen must, in particular, be capable of modelling the tensile failure of the bricks as observed in post-earthquake damage reports [32,33]. This approach is used to analyse small structural elements such as a specimen of masonry shear triplets [34,35]. In this paper, an extensive experimental study was conducted on: (i) non-reinforced masonry triplet specimens; and (ii) strengthened masonry triplets with carbon fiber-reinforced polymer (CFRP). The masonry triplets were constructed with two types of mortar compounds and tested at three different levels of axial pre-compression to evaluate the main brick–mortar interface characteristics. A numerical model was carried out to assess the main shear response of masonry triplets and the main failure mode by considering the shear slip and the normal separation at the brick–mortar interface using the detailed micro-modelling (DMM) approach. In addition to this, an extended finite element method (XFEM) was used to simulate the crack initiation and propagation in the mortar layer. The results of these numerical examples were compared with the experimental results.

## 2. Materials and Methods

### 2.1. Materials

#### 2.1.1. Brick Units

The masonry units used to construct the triplet prisms required in this research present nominal dimensions of 220 × 105 × 55 mm^3^. The bricks had a density of 1800 kg/m^3^. A series of five tests have been studied to obtain the average values of compressive strength and the elastic modulus. The tests were carried out using a hydraulic press according to the EN 771-1 [36]. In further accordance to Vasconcelos and Lourenço [37], the$\;E_{b}$ is calculated by considering values between 30% and 60% of the compressive strength, resulting in an average Young’s modulus of 10,000.13 MPa and average compressive strength of fb = 24.24 MPa, respectively, with a coefficient of variation (CV) of 10.3%. The tensile strength of perforated bricks was indirectly measured by means of the three-point load test (see Figure 1). The three-point load tests were applied to three equal bricks, obtaining an average flexural strength based on the gross area of ft = 1.453 N/mm^2^ with a CV of 12.6 %. N/mm^2^. The mechanical properties of bricks are measured and presented in Table 1.

#### 2.1.2. Mortar

Two weight batching mixes of cement, lime, and sand (1:1:3 and 1:1:5) were prepared using an electrical mixer. To achieve a workable consistency, water was added, and the mixture was remixed. The prepared mixtures were cast into standard molds and then maintained in the standard curing to measure the mechanical properties of these mortars.

Mortar mechanical properties were determined at 7 and 28 days after de-molding. EN1015–11 [38] was used to conduct flexural and compressive strength tests. Flexural strength was tested on prism-shaped specimens (40 × 40 × 160 mm^3^) using a universal testing machine (see Figure 1). The two half-prisms obtained after breaking the specimen into two parts during the flexural test were then subjected to the uniaxial compressive test. The average value of the response is determined by testing three specimens of each mix. The mechanical properties of the mortars were experimentally measured and reported in Table 2.

#### 2.1.3. CFRP Strips

All the specimens were strengthened by applying one layer of unidirectional CFRP (Sika Wrap carbon fiber fabric) strips with a thickness of 2.5 mm and a width of 15 mm. According to technical data provided by the producer, the strip had a minimum tensile strength of 3100 MPa. The measured mean value of elastic modulus was 165,000 MPa. Strips were glued using a commercial two-component epoxy resin; its tensile strength and elastic modulus were 30 MPa and 4800 MPa, respectively. The properties of the CFRP reinforcing system are presented in Table 3.

### 2.2. Shear Triplet Tests Setup and Procedure

This section describes the specimen configuration, conditioning, and testing methods employed to assess the shear strength of masonry triplets under non-strengthened and strengthened conditions. In this test, the specimens are composed of three perforated bricks that are attached together by two 10 mm-thick mortar joints, as shown in Figure 2, leading to masonry triplets with dimensions of 210 × 185 × 105 mm^3^. Before making the specimens, all bricks were immersed in water for 24 h to ensure a better adhesion with the mortar joints and to avoid rapid drying of the specimens. After that, the triplet specimens were left to cure for 28 days in the laboratory at a temperature of about 25 °C.

In order to obtain a pure shear state in the joints, a shifted triplet specimen with a fixed length of mortar joints is proposed to be utilised. The difference between the triplet test set-up proposed here and the standard triplet is that the outside header of the middle brick is helpful in applying a shear force. Crisafulli [39] adopted this test configuration to study the shear properties of the mortar joints. These specimen prisms are placed longitudinally under the load that applies to the head of the middle brick masonry unit. This experiment can be done with or without lateral pre-compression load, but for the determination of the mechanical parameters of the brick–mortar interface, such as friction coefficient, it is required to achieve both experimental tests. To assess the initial strength and friction angle of the unit–mortar interface, four levels of normal compressive stress (0, 0.2, 0.6, and 1.0 MPa) were used in accordance with BS EN1052-3 [40]. There is a requirement for an adequate test setup for applying constant pre-compression while applying shear force. For this reason, it is necessary to use a novel design specifically fabricated for carrying out this test.

An independent system was used to apply the horizontal compressive stress on masonry triplets. This system setup was fabricated to accommodate different pre-compressive levels of normal stress for shear triplet specimens; it consists of two steel plates placed at the two sides of the triplet and assembled by two threaded rods. In the beginning, for each specimen, the required confining pressure was applied to the steel plate system by means of a mechanical actuator, which remained constant throughout the test, before gradually applying a shearing load as compression. The maximum loading capacity of the vertical actuator is 500 kN. The shear load was applied at a rate of 0.05 mm, and the corresponding shear displacement was measured by means of four LVDTs attached to two opposite sides of the specimen and recorded through a data logger almost constantly throughout the entire loading process. According to the testing programme, for each mortar composition (1:1:3, 1:1:5), three different samples were tested for each of the four levels of normal compressive stress (0,0.2, 0.6, and 1 MPa), for a total of twelve masonry triplets tested. Then, similarly for the reinforced shear tests, four masonry triplets were performed for two configurations of reinforcement.

Two different reinforcing patterns of shear triplet specimens with CFRP based on failure modes are presented in Figure 2, which can be applied to answer such types of failure mechanisms. It shall be noted that the CFRP strips with angles of 0 and 45 degrees are used on one side and both sides. The CFRP reinforcement process was performed after seven days of the construction of triplets. The surfaces were treated before applying the CFRP. To apply this strips, the face sides of the masonry triplets were cleaned with sandpaper to remove any excessive hardened mortar from the joints and loose particles on the surface. A thick layer of two component saturating epoxy resin was applied on the masonry surface using paint roller. The reinforced masonry triplets were strengthened by SIKA Warp CFRP of 15 mm width and 2.5 mm thickness. The CFRP strips were bonded using a wet lay-up process in the vertical direction, parallel to the mortar joint, as well as in the diagonal of the specimen at 45° to the bed joints. The curing period of reinforced triplet specimens was at least 7 days prior to testing. The experimental setup adopted to perform the shear tests on masonry triplets is shown in Figure 3.

The CFRP-reinforced shear triplet was subjected to the same pure shear loading. The specimen reference adopts the format STxx (A) or STxx (B), (A and B), which denotes masonry triplets made with the mortars 1:1:3 type A and 1:1:5 type B, respectively; xx represents the pre-compression stress (e.g., 02 for 0.2 MPa, and 10 for 1 MPa); (STC) refers to the control triplet specimen, (SRV) denotes a triplet specimen reinforced with vertical CFRP strips on both sides, (SRX) indicates a diagonal CFRP reinforcement applied to one side of the masonry triplet specimen, and (SR2X) indicates the specimen reinforced with diagonal CFRP on two sides.

The behaviour of the shear triplets was studied by calculating the shear strength after finding the ratio between the load and the net area parallel to the mortar joint, as follows:(1)$$\tau =\frac{P_{\max}}{2A}$$

*P*_max_: the shear load at failure and *A* is the total cross-sectional contact area between two bricks.

## 3. Results and Discussion

### 3.1. Failure Patterns

Figure 4 summaries the failure modes observed for all specimens under different lateral confinement stresses. From these results, it is clear that the shear triplet prisms constructed with mortar type A provide a similar failure mode to the shear triplet prisms type B. This means that no relationship was observed between the type of failure and materials.

Specimens under null confinement stress presented a rupture mode which occurred by slipping along both brick–mortar interfaces and shearing crack in the mortar layer. In this case, the failure modes were predominantly caused by mortar layer failures. During the increase of the shear load, a diagonal crack was observed in the mortar layer and propagated along the mortar, therefore, leading to the specimen being separated into two distinct bodies from either the inner brick or the outer, or both. No substantial damages were seen on the brick side (Figure 4a,c). This may be attributed to the lack of frictional resistance due to the absence of compressive stress normal to the bed joint during the shearing stage. A similar failure mode was likewise reported by Prakash [41] and Bompa et al. [42]. When increasing the confinement stress level, the observed failure mode was a combination of sliding along the mortar–brick interfaces and shear cracks appeared near the interface through the mortar layer (see Figure 4d). During the increase of the shear load, a minor crack was observed, propagating into the outer bricks at the peak shear load. Cracking and crushing of the bricks were also observed at the end of the tests in some cases (Figure 4b,d). All specimens failed in the brick–mortar interface either on one or divided between two brick faces, which are types of failure corresponding to failure modes described in [40].

Figure 5 demonstrates the relationship between the shear strength and the normal stress of shear tests performed on masonry triplets made with the mortars 1:1:3 type A and 1:1:5 type B, respectively. It can be observed in this figure that the shear strength increases with both confining pressure and mortar strength. Similar findings were also reported by Abdou et al. [43]. It is observed that the shear triplet presented an approximately linear shear stress–strain relationship before cracking, followed by a sudden drop of shear strength once cracking had propagated in the mortar layer, followed by a slight increase in shear strength and deformation with the application of further loading before rupture. This occurred when cracks appeared in the mortar layer and interface of the triplet. There is a barely detectable hardening phase before the peak load but just for the specimen type A.

According to Figure 5a, it is easy to see that just before the peak shear stress the stiffness is very high with very little shear deformation. The interlocking between the grains of the brick and the mortar under confining pressure is the main reason for the high stiffness of the shear stress–displacement relationship.

Figure 5b shows that the interface behaves like a quasi-brittle material, with a very small hardening branch appearing between the elastic limit and the peak stress. Following the initial damage, the post-peak damage and release of strain energy are quite visible as the stress gradually decreases. In terms of initial stiffness, all specimens demonstrated a rigid response up to the maximum shear stress, with the exception of the specimens under null compression. This demonstrated a brittle failure mode, resulting in a sudden drop of the curve due to the absence of the confinement stress. The gradual degradation of the interface shear response after reaching the peak is clearly visible.

### 3.2. Joint Strength Parameters (Joint Shear Test)

The interface shear behaviour of masonry is important in developing predictive approaches for masonry wall in-plane shear response. In such walls, the failure is most likely to occur at the brick–mortar interface, resulting in separation of the two materials. The frictional resistance would continue to transmit shear stress through the separated elements. The bond or adhesion between unit and mortar is often the weakest link in masonry, as is the characterisation of the unit–mortar interface. According to Lourenco [44], at the unit–mortar interface, two distinct phenomena can occur: one associated with tensile failure (mode I) and the other with shear failure (mode II). In order to characterise the shear bond behaviour, triplet tests were carried out under different normal compressive stresses. The failure of the unit–mortar interface of masonry under shear can be characterised by the Coulomb friction failure envelop for lower levels of normal compressive stress to the joint (2 MPa).
(2)$$\tau_{u}=C+\mu *\sigma_{n}$$

μ=tanφ: coefficient of friction between the unit and the mortar

$C=\tau_{0}$: the shear strength under zero compression loads that represents the cohesion (bond shear stress).

$\sigma_{n}$= $\sigma_{0}$: the normal stress

In this study, two important parameters, confining pressure and mortar strength, were noticed as major factors contributing to the shear capacity at the brick–mortar interface. The experimental parameters found from the shear triplet tests are given in Table 4.

From Table 4, it appears that the ultimate shear strength is strongly affected by the pre-compression stress with a limited effect of the mortar compositions. The ultimate shear strength of STA and STB shear triplets under zero compression loads were measured as 0.71 and 0.44, respectively. From the results analysis, it was shown that the ultimate shear strength of the specimen increases as the cement-to-sand ratio in the mortar increases. These results indicate that the ultimate shear strength increases with increasing confining pressure normal to the shearing surface for both types of shear triplets.

It is quite evident from Figure 6a that there is an almost linear relationship between the shear and normal stresses (*σ*-*τ*), as represented by the linear relationships:(3)$$\tau =0.57\sigma_{n}+0.762$$
(4)$$\tau =1.04\sigma_{n}+0.435$$

According to this formulation, for the two types of mortar (A and B) the value of cohesion C (the initial shear strength of masonry) was 0.762 MPa and 0.435 MPa, respectively. The value of the coefficient of friction (*μ*) equals 0.57 and 1.04, respectively. It was noted that the values of cohesion and internal friction angle obtained here are within the range reported in the literature [45], so this author suggested an average cohesion equal to 3% of the compressive strength, and 0.3 to 1.2 for the coefficient of friction of the masonry.

Figure 6a also shows that the strength of the mortar plays an important role in the peak shear stress. The other two parameters, cohesion (c) and internal friction angle, which are inherent properties of the brick–mortar interface, also vary with mortar strength. The cohesion c is independent of normal stress and decreases as mortar strength decreases, whereas the coefficient of friction increases as mortar strength increases. The interface cohesion c value represents the strength of the bond between the brick and mortar interfaces. However, increased mortar strength is not the only governing factor that influences the shear strength of the brick–mortar interface. Other elements include brick and mortar characteristics, surface roughness and strength of the brick, bond characteristics between mortar and brick, and overall joint quality. As overall uniformity was difficult to achieve during specimen fabrication, some inconsistencies are inevitable.

### 3.3. CFRP-Reinforced Shear Triplets

Figure 7 shows the failure mode of all reinforced specimens. The failure in the specimen reinforced with horizontal CFRP, as shown in this figure, occurred through the development of sliding on the brick–mortar interface and starting of shear cracks in the mortar layer, whilst the specimen reinforced with diagonal CFRP strips on both sides was more ductile than the control specimen, but it failed due to the same failure mode. Moreover, the CFRP strips also debonded when this specimen failed completely.

The curves shown in Figure 8 reveal remarkable differences in the ultimate shear stress when maximum stress was reached. As shown in Figure 8, the shear stress vs. shear displacement curve of reinforced triplets with one side of CFRP portrayed a more sudden drop in stress after the peak was reached. The specimen reinforced by two vertical CFRP strips (115RV) extended along the mortar joint shows brittle failure with a sudden loss of strength, whereas the specimen (115R2X) reinforced by diagonal CFRP strips on both sides had less brittle failure but a larger deformation, with the latter developing a more gradual post-peak softening.

Although the strength increase due to the presence of vertical CFRP strips was modest, the post-peak performance was more favourable, consisting of a more gradual stress reduction with the increase in slip.

Table 5 indicates that all the reinforced specimens revealed a significant improvement in ductility compared with the corresponding control specimen. Similar results were also reported in the literature by Saghafi et al. [24]. The results obtained from this table show that the reinforcement with CFRP strips parallel to the joint (vertical) improved the shear strength by almost 7 to 14%, whereas CFRP reinforcement on both sides in an X pattern increased the shear strength by almost 107 to 150%. Ductile behaviour was also obtained, and the deformation capacity of the specimens increased by 47 to 90%.

### 3.4. Numerical Modelling and Set-Up

In this section, the detailed micro-modelling (DMM) approach is adopted for simulating the shear behaviour and failure mechanisms of the tested shear triplet prisms based on the behaviour of the basic constituents, presented in the previous section. Brick and mortar joints are separately modelled by concrete damage plasticity model (CDP) constitutive laws. To capture the behaviour of the bond between the FRP strip and the masonry layer, especially for those that debonded, an interface model was chosen in order to accurately model the masonry wall strengthened with FRP composite. The debonding in the CFRP-strengthened masonry interface was modelled using the cohesive surface available in ABAQUS/implicit.

The units and mortar joints were modelled using eight-node 3D continuum elements with four glass controls and reduced integration (C3D8R). The unit–mortar interface was modelled as a cohesive interface with zero thickness to simulate the failure mechanisms at the interface (tangential sliding and normal separation) and the XFEM was utilised to simulate the crack progressions and the crack initiation at the mortar layer level. To model cohesive behaviour, the normal and tangential stiffness of the traction and separation law in kn and ks was introduced. After carrying out the mesh convergence studies, on a scale of accuracy, a uniform mesh size of 2 mm was used for the prism modelling. Boundary conditions have been selected to match the experimental setup constraints. For the detailed micro-model of shear triplets, the incremental compressive load was applied at the top surface of the middle brick in terms of displacement. The bottoms of the prism, such as the bottoms of right and left units, are restrained in the directions against the loading. The CFRP strips were modelled in ABAQUS using the lamina material properties and 3D shell elements; they were modelled as a linear material orthotropic. The geometry of assemblies, the loading condition, and defined interaction surfaces between the units and mortar are shown in Figure 9.

#### 3.4.1. Mechanical Model for Brick and Mortar

The concrete damage plasticity model (CDP) used here to identify the behaviour of units and mortar was developed by Lubliner et al. [46] and is available in ABAQUS. It was used here to simulate the nonlinear behaviour of masonry constituents and to predict the two main types of failure modes: cracking in tension and crushing in compression. The level of damage is represented by the parameters *dc* and *dt* as defined by Wang et al. [21].

The brick and mortar units were modelled by first using the elastic modulus and Poisson’s ratio, and then the damage plasticity model to define the nonlinear part of the stress–strain curve. Further details can be found in Ref. [9], where in this study, the curves adopted for the plastic behaviour in compression and in tension and the evolution of the scalar damage variables were carried out.

There are other parameters needed for applying the CDP technique in ABAQUS:The dilation angle(s) (ψ) is 12° to 30°. In this study, the dilation angle was 20°.The eccentricity parameter (e) ranges from 0 to 0.1 and is assumed to be 0.1.Viscosity parameter = 0.001.($\frac{f_{bo}}{f_{co}}$) the ratio between the initial equi-biaxial compressive strength and uni-axial compressive strength of masonry when the default value was used (1.16).(*K*) the ratio of the second stress invariant on the tensile meridian when the default value was used (0.67).


#### 3.4.2. Interface Properties

The cohesive behaviour of the surface and the traction–separation law were used to model the interface between the units and mortar with zero thickness. In ABAQUS, the traction separation model has three criteria: linear elastic behaviour, the damage initiation criterion, and the damage evolution law [47]. Before there is any damage, the initial response of the joint interfaces has a linear traction–separation relationship as developed by Lubliner et al. [46]. In the elastic part, the general linear behaviour is defined according to the relationship between the nominal traction stress (*t*) and nominal strain (*δ*) through the interface.

The relationship between the elastic stiffness matrix (*K*), the traction stress vector (*t*), and the separation vector (*δ*) through the interface can be expressed in standard form as in Equation (5). The components (*t_n_*, *t_s_* and *t_t_*) represent the fracture modes.
(5)$$t=\left(\begin{array}{c} t_{n}\\ t_{s}\\ t_{t} \end{array}\right)=\;\left[\begin{array}{c} K_{nn}0\;\;\;\;0\\ 0\;\;\;\;K_{ss}\;0\\ 0\;\;\;\;0\;\;K_{tt} \end{array}\right]\left\{\begin{array}{c} \delta_{n}\\ \delta_{s}\\ \delta_{t} \end{array}\right\}\;\;$$

*n* and *s* are the normal and shear components, respectively.

Uncoupled cohesion has three stiffness components in normal (*K_nn_*), and the two-local shear (*K_ss_*) and (*K_tt_*) directions are considered.

The equivalent stiffness for joint interfaces is represented as a function of the unit and mortar elastic modulus and the thickness of the mortar (see Equations (6) and (7)) [44].
(6)$$k_{n}=\frac{E_{u}E_{m}}{h_{m}\left(E_{u}-E_{m}\right)}\;\;\;$$
(7)$$k_{s}=k_{t}=\frac{G_{u}G_{m}}{h_{m}\left(G_{u}-G_{m}\right)}\;\;\;$$

*h_m_* is the thickness of the mortar

*E_u_* and *E_m_* are Young’s moduli for unit and mortar, respectively.

*G_u_* and *G_m_* are the shear moduli for mortar and unit, respectively.

The second part of the traction–separation response shows the damage propagation of the bond. When the maximum nominal stress ratio equals one, the damage is initiated. Cohesive elements are used to bond two parts and they degrade when a load due to the tensile or shear deformation was applied. Afterwards, the two bonded parts reach into contact after debonding. The damage evolution of cohesive behaviour represents the progressive degradation of cohesive stiffness or the dissipation of energy; it is called fracture energy and depends on the fracture mode that resulted from the failure method. The opening of the interface in a normal direction is called mode I, and the second and third modes are shearing modes known as mode II and mode III. In addition, it is necessary to avoid element penetration after making the contact, especially for the normal behaviour of contacts. For this reason, contacting properties for the tangential and normal behaviour were specified in this work.

#### 3.4.3. Modelling of CFRP and Masonry to CFRP Bond Interface

The linear elastic response of CFRP is defined by using the lamina model needed to define the elastic modulus, the shear modulus in two directions, and Poisson’s ratio. The Shell elements S4R are utilised to represent the CFRP because there are six degrees of freedom per node. The debonding of CFRP and the contact region between CFRP and the masonry surface was modelled using surface-based cohesive behaviour. The interface was modelled using the cohesive zone. The bond between the composite and the masonry was modelled with contact cohesive behaviour to represent the masonry–composite interface with the initial stiffness presented in Equation (8) [47].
(8)$$k_{0}=\frac{1}{\frac{t_{i}}{G_{i}}+\frac{t_{c}}{G_{c}}}$$
where *t**_i_* is the thickness of the resin, *t_c_* is the thickness of the masonry wall, and *G_i_*, *G_c_* is the shear modulus of the resin and masonry wall, respectively.

Two criteria were utilised to evaluate the initial debonding of the CFRP–masonry interface. One criterion assumed that the mode I and mode II debonding was independent, while the other criterion assumed that normal and shear stress have couplet effects and thus mixed-mode interface debonding will begin when the following stress condition is reached [48]. The quadratic function was used to indicate the initial damage of the interface; it is presented in Equation (9):(9)$$\;\;\;\;\;\;\;\;\;\;\;\;\;\;\;\;\;\;\;\;\;\;{\left(\frac{\sigma_{n}}{\sigma_{n}^{0}}\right)}^{2}+{\left(\frac{\tau_{n}}{\tau_{s}^{0}}\right)}^{2}+{\left(\frac{\tau_{t}}{\tau_{t}^{0}}\right)}^{2}\;\{\begin{array}{c} <1\;no\;failure\\ =1\;failure \end{array}$$

*σ_n_* is the tensile stress, *τ_s_* and *τ_t_* are the shear stresses of the interface, and *n*, *s*, *t* are the directions of the constraint components. The values used for epoxy resin are $\sigma_{n}^{0}$ = 1.81 MPa $\tau_{s}^{0}$ = $\tau_{t}^{0}$ = 1.5 MPa [49].

The results obtained from characterising the materials in this study were admitted (values were *fc*, *c*, and *μ*). The dilatancy angle was assumed to be zero to avoid predicting non-conservative shear strength.

For the tensile fracture energy (mode I) of the interface, the data available in the literature (Pluijm) recommends values from 0.005 to 0.2 N/mm for a range of tensile strengths from 0.3 to 0.9 N/mm^2^. This was confirmed by Barros and al. [50] when they found that for different types of brick and mortar interfaces, the average mode (I) fracture energy was around 0.008 N/mm when the average bond tensile strength was in the order of 2 N/mm^2^. The shear tractions were known as the initial shear strength, it is related to the cohesion parameter (c) at zero confining stress. It is well known that this value depends on the mechanical properties of masonry assemblage and the amount of the applied vertical load. The masonry shear triplets were modelled according to the materials’ properties reported in Table 6 and Table 7.

### 3.5. Numerical Results for Finite Element Modelling and Comparison with Experimental Results

In order to evaluate the effectiveness of this proposed approach, the results from experimental tests conducted in this research were compared with those obtained from the developed numerical model, in terms of shear stress–displacement curves and failure modes.

Figure 10 shows the damage and the STUXFEM of unreinforced shear triplets at different levels of pre-compressive normal stress. This figure shows that for specimens without pre-compression, the failure was characterised by the commencement of slipping on the brick–mortar interface. In addition, a diagonal cracking occurred at the middle of the mortar layer, therefore, leading to the specimen being separated into two distinct bodies, with the shear stress dropping instantaneously. On the other hand, for specimens tested with pre-compression stress, the failure occurred through the development of the slipping and the appearance of a micro-crack near the interface which further propagated through the mortar layer. Moreover, this figure shows a comparison between the failure modes developed in the numerical model and experimental test, with pre-compression stresses of 0, 0.2, 0.6, and 1 MPa, respectively. It can be observed from Figure 10a that the tensile damage at the bottom of the right-hand brick appeared and increased when the pre-compressive stress increased. This finding has been also observed by Beattie and al [51]. Similar findings were also reported in the literature by Abdou et al. [43], Fouchal [52], and numerically by Sarhosis and Lemos [53].

The crack patterns observed in the mortar layer and the brick during the experimental test and those predicted in the proposed model were similar. This means that the numerical model can capture the failure modes of shear triplets with sufficient accuracy.

This is in agreement with EN 1052-3 [40], which states that the diagonal crack of the mortar joints and slipping at the brick–mortar interface are the most common failure mechanisms in masonry assemblages under shear stress.

The shear stress distribution and the STUXFEM for all the reinforced specimens are shown in Figure 11. It can be identified that the shear failure of this masonry specimen can be eliminated by CFRP reinforcement, which also prevented the micro-cracks from extending through the mortar layer. This proves that CFRP reinforcement helped to support the tensile force at the bed joints. Furthermore, as shown in Figure 11, the debonding of the CFRP composite was predicted by the contact opening parameter (COPEN), and contact shear (CSHEAR). This figure shows a comparison between the failure modes developed in the numerical model and experimental test. It can be seen from the figure that when the specimen was reinforced with vertical strips, the debonding appeared at the end of the strips, but there was no debonding along the CFRP strips.

On the other hand, for the specimen reinforced with diagonal strips, the debonding appeared in the edge of the four sides of the strips and merely before the intersection of the strips. As shown in this figure, a similar failure mechanism was likewise reported in the experimental test of this study. In fact, according to these parameters, the proposed contact models performed well enough to capture the location of debonding of the CFRP composites. However, there is some position of numerical debonding of CFRP reinforcing that differs from the experimental test. This is exemplified in the diagonally reinforced specimen, which can be explained by inadequate implementation of the method test for connecting the CFRP to masonry interface. Among them, preparation and brushing of a surface of contact before the application of epoxy resin required more skilled labor for this purpose, which is not assured experimentally, causing a significant effect on the overall structural performance of the CFRP composite. All these results demonstrated that the numerical analyses provide helpful insight into phenomena that are virtually difficult to observe experimentally.

As depicted in Figure 12a–c, the numerical curve shows more stiffness than the experimental curve and the load-bearing capacity indicated 3 and 9 percent error for the specimens under pre-compression of 0, 0.2, and 0.6 MPa. Furthermore, the percentage error of the ultimate displacement ranged between 1 to 7, whilst for the specimen under normal stress of 1 MPa, the percentage error of stress and ultimate displacement reached the values of 26 and 49, respectively (see Table 8). The ultimate displacement obtained in the experimental test was less than predicted by the numerical model, as Table 9 indicates. Moreover, the numerical model for the diagonally reinforced shear triplets overestimated the shear strength by less than 21%, possibly due to the bond between the CFRP and masonry surface. During the fabrication of the specimens, an overall uniformity was difficult to attain, therefore, some inconsistencies are inevitable.

Overall, all the stress versus displacement curves show good accordance in numerical and experimental results, and it can be stated that the models are well-calibrated. Moreover, the adopted numerical approach appears satisfactorily accurate to investigate the debonding phenomenon in FRP-strengthened masonry.

## 4. Conclusions

An experimental test program and numerical simulation was carried out to study the shear behaviour and failure mechanism of brick masonry triplets reinforced with CFRP strips under different levels of pre-compression and two different mortar mix ratios. A detailed micro-modelling strategy was adopted. The failure mode and the stress–displacement curves obtained from the experimental test were compared to the numerical results. The proposed model can effectively predict the peak shear stress as well as the initial stiffness and the failure pattern in the mortar layer and the interface for the experimental results. Based on the results obtained from this study, the main conclusions are summarised in the following points.

The strength of the mortar has a limited effect on the peak shear stress. The other two parameters, namely cohesion (c) and internal friction angle (ϕ), also vary with the mortar strength. On the other hand, higher confining pressure levels increased the shear capacity of the interface.The cohesion (c) is independent of normal stress and decreased with decreasing mortar strength, whereas the coefficient of friction increased with increasing mortar strength.Beyond the shear response, the triplet failure modes were sensibly affected by the compression level, independent from the mortar joint properties.The tensile damage at the bottom of the brick largely depended on the applied level of pre-compression stress.For reinforced shear triplets, the failure was governed by the brick and mortar tensile properties rather than that of CFRP strips, due to their relatively high strength.Specimens provided with diagonal reinforcement exhibited a higher strength enhancement than those with vertical reinforcement for two types of specimens. In this case, the diagonal configuration in both types of mortar proved to be effective in restricting brick–mortar interface separation and maintaining specimen integrity.A significantly higher contribution from CFRP was observed regarding strengthened shear triplets, for which the strength increase provided by CFRP was in the range of 107 to 150%, depending on the reinforcement configuration.The significant increase in ductility and shear capacity was achieved by diagonal CFRP reinforcement on both sides of the masonry specimen.The extended finite element method (XFEM) was suitable for modelling crack propagation in the mortar layer. It also helped to recognise the region where cracking occurred in the mortar layer and where the specimen failed.

Based on the experimental results and developed numerical model of this paper, various studies are to be carried out in the future to propose an interface model which predicts the exact failure mode of brick masonry walls. In addition, an explicit correlation between friction coefficient, mortar strength, and brick strength should be established.

## Figures and Tables

**Figure 1 polymers-14-03707-f001:**
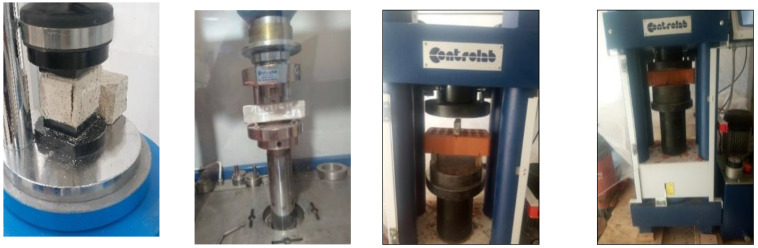
Three-point bending test and compressive test for mortar prisms and brick specimens (Compressive strength machine, Controlab, Saint Ouen L’Aumône, France).

**Figure 2 polymers-14-03707-f002:**
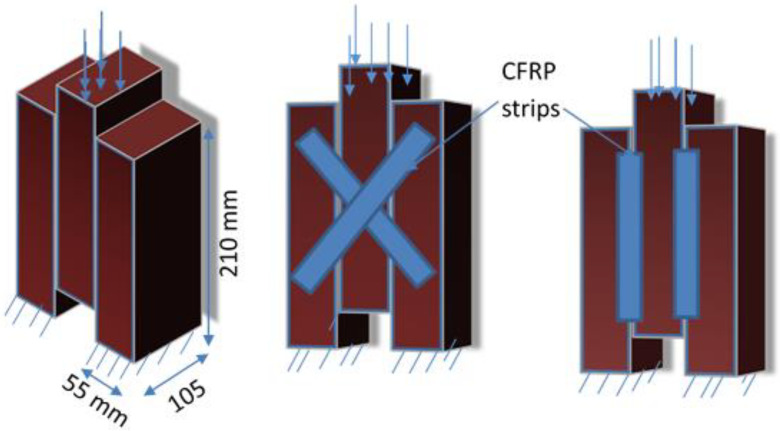
Masonry samples geometry (staggered-triplet specimen).

**Figure 3 polymers-14-03707-f003:**
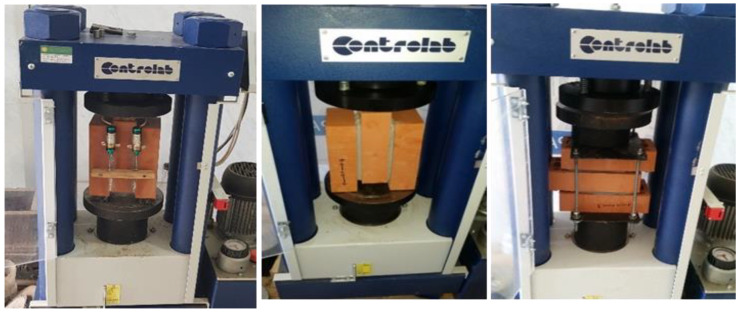
Experimental setup of masonry shear triplets.

**Figure 4 polymers-14-03707-f004:**
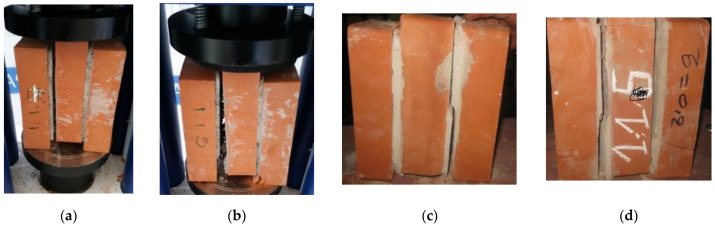
Typical failure modes observed during the shear triplet tests: (**a**) sliding a long bed joint, (**b**) diagonal shear cracking at mortar and crashing at the bottom of right, (**c**) diagonal shear crack at mortar on the right side, (**d**) diagonal shear cracking at mortar and sliding at the brick–mortar interfaces; cracking appears at the bottom of right brick.

**Figure 5 polymers-14-03707-f005:**
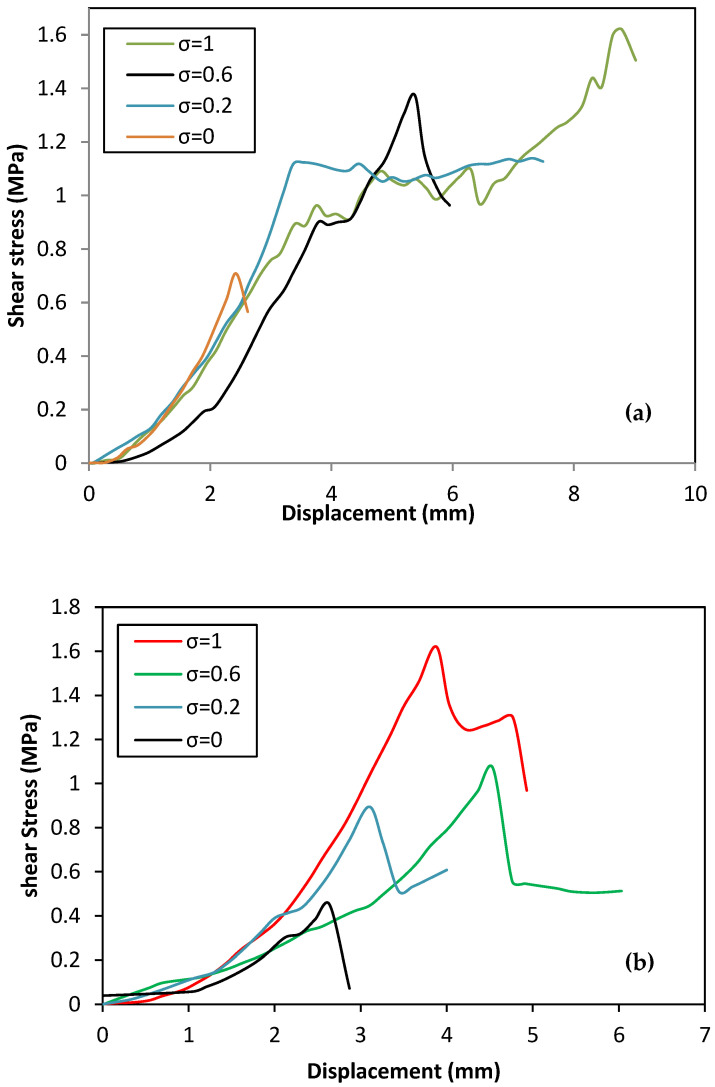
Stress-strain curves for various pre-compression levels for non-strengthened triplets: (**a**) STA, (**b**) STB.

**Figure 6 polymers-14-03707-f006:**
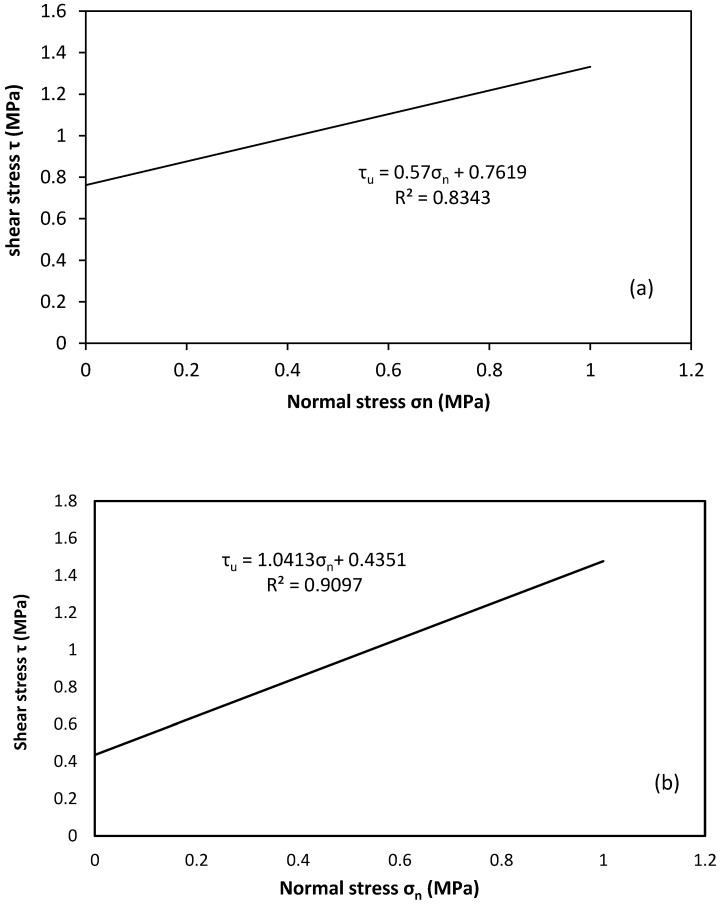
Maximum shear strength versus pre-compression level for non-strengthened shear triplet test: (**a**) STA, (**b**) STB.

**Figure 7 polymers-14-03707-f007:**
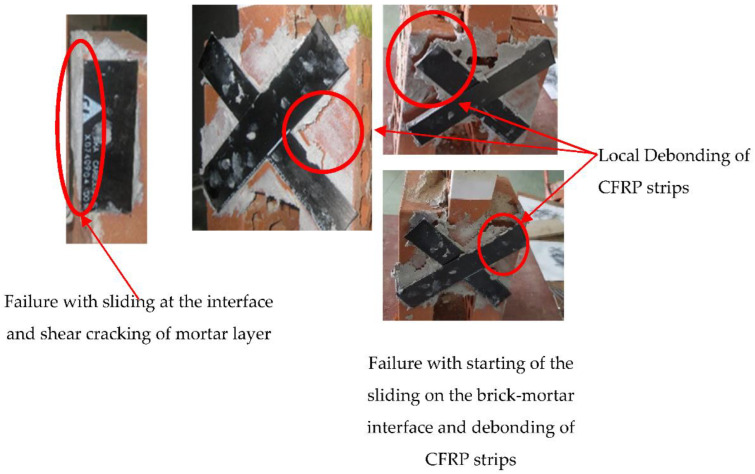
Failure mechanisms of reinforced masonry triplets.

**Figure 8 polymers-14-03707-f008:**
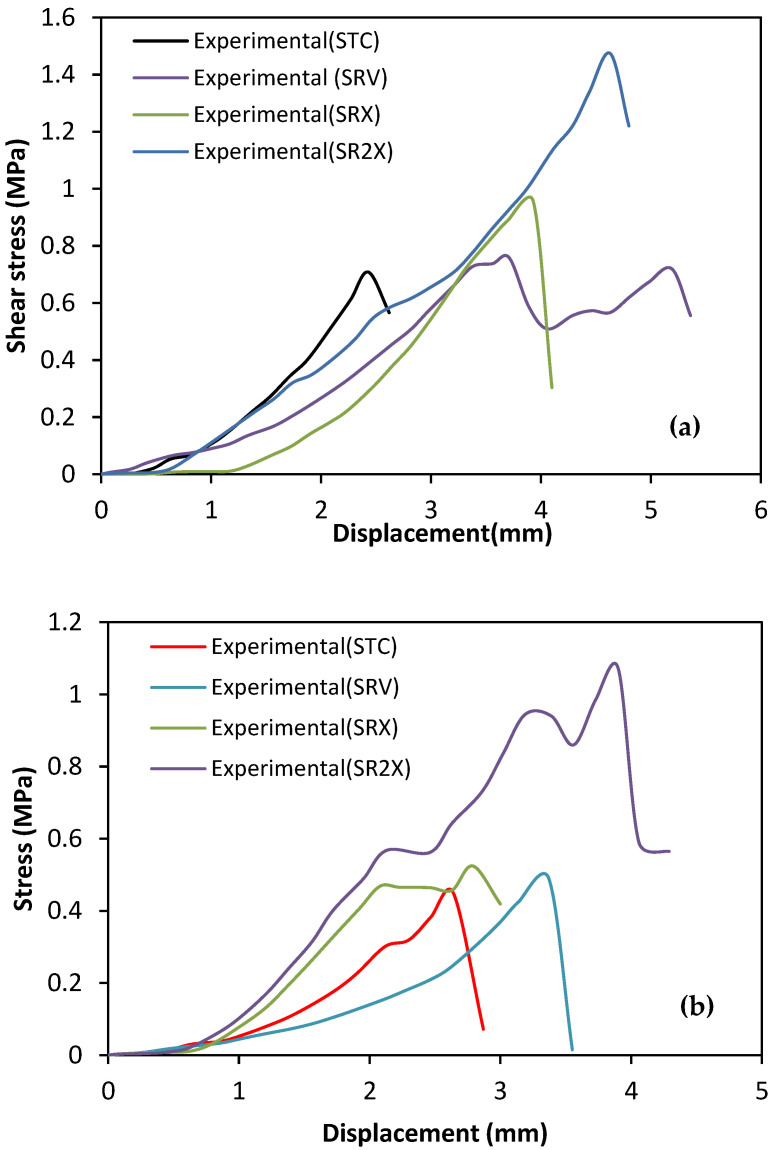
Stress–strain curve for unreinforced and reinforced masonry triplets with different disposition of CFRP strips: (**a**) masonry triplets constructed with mortar A; (**b**) masonry triplets constructed with mortar B.

**Figure 9 polymers-14-03707-f009:**
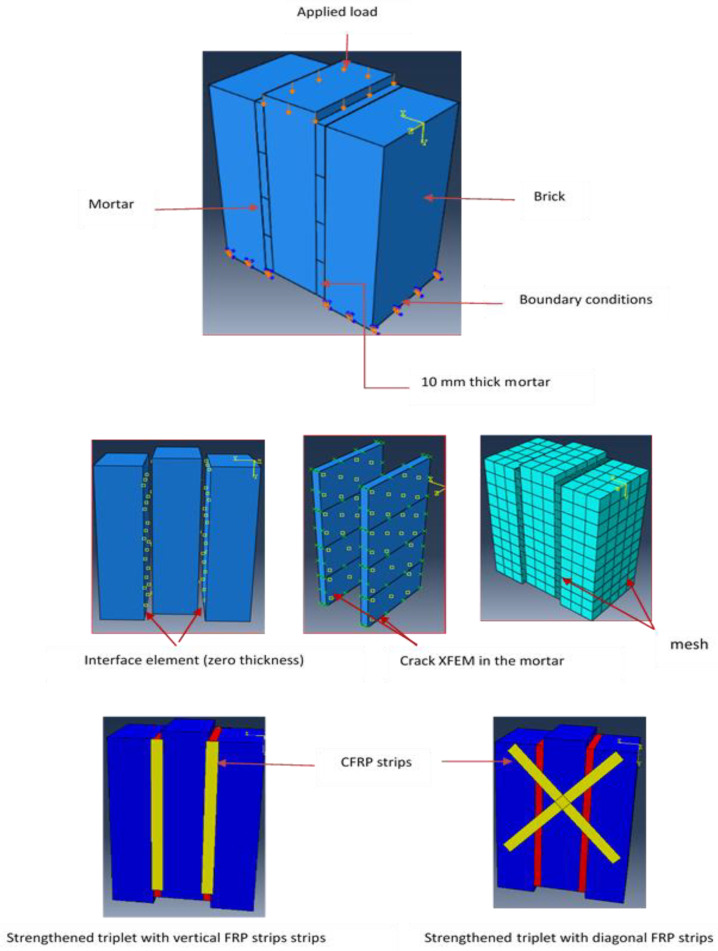
Triplet model with loads and boundary conditions.

**Figure 10 polymers-14-03707-f010:**
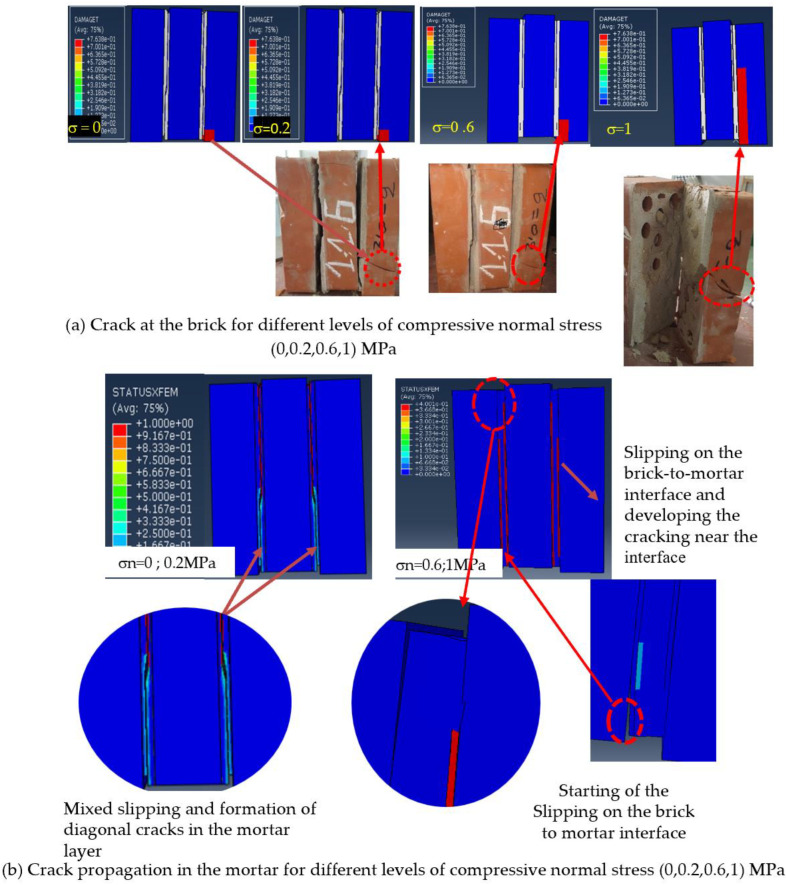
Failure mechanisms of unreinforced masonry shear triplets under different levels of pre-compression obtained from experimental tests and the numerical model (**a**) for brick; (**b**) for mortar.

**Figure 11 polymers-14-03707-f011:**
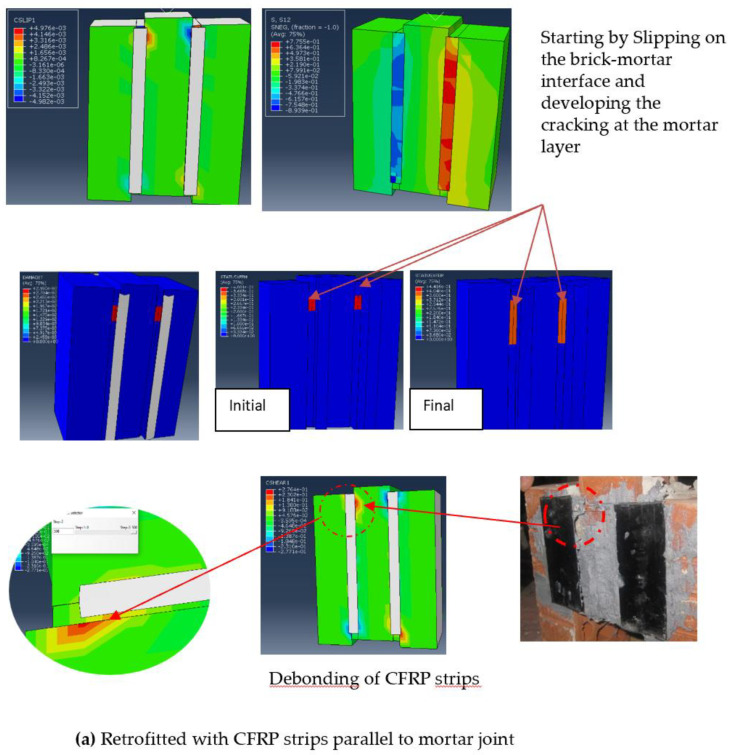
Shear stress contour plots; failure mechanisms for reinforced masonry shear triplets with different configuration of FRP strips: (**a**) retrofitted with CFRP strips parallel to mortar; (**b**) retrofitted with diagonal CFRP strips (X form pattern).

**Figure 12 polymers-14-03707-f012:**
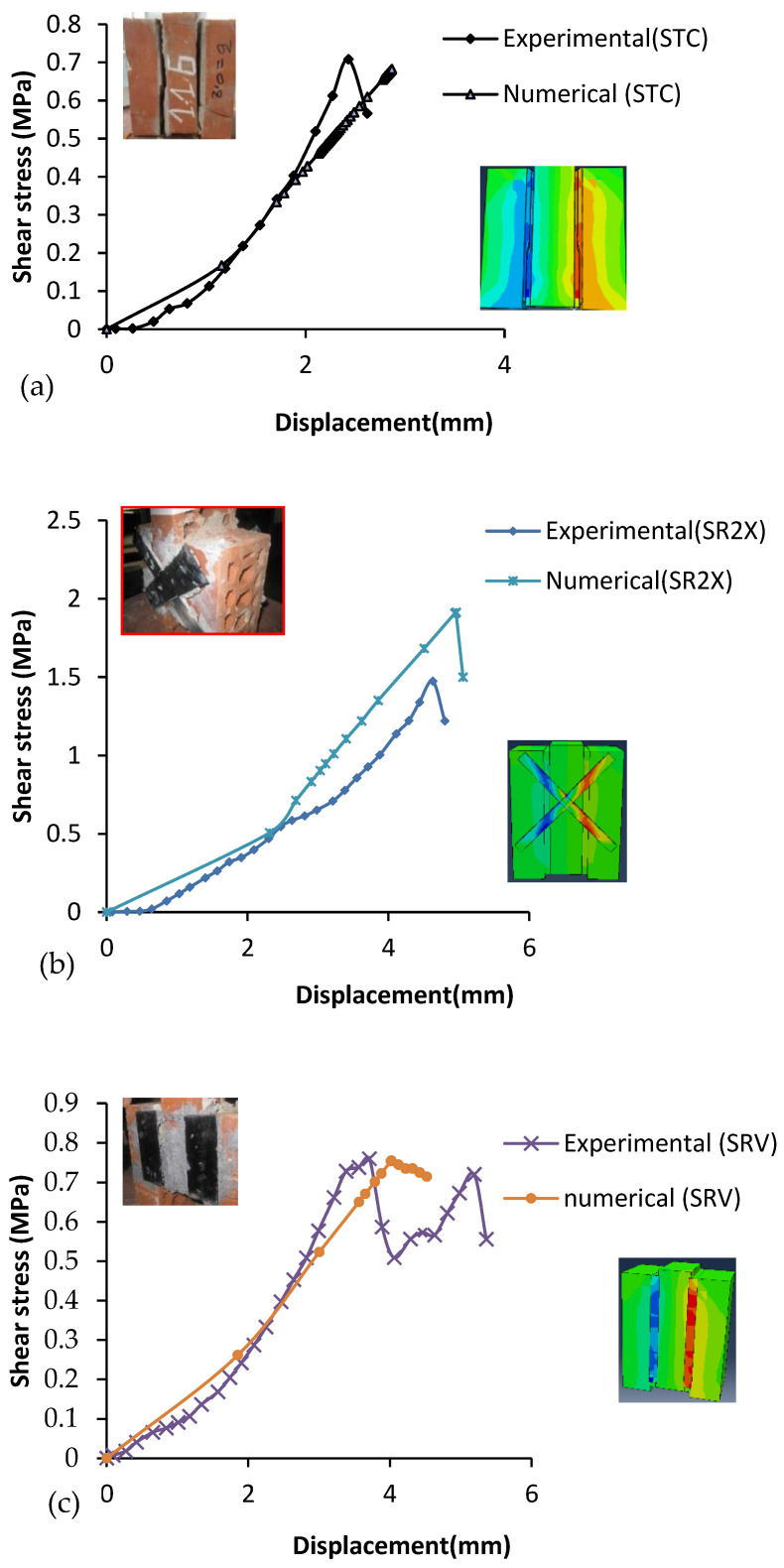
Comparative assessment between numerical results and experimental test of non-reinforced and reinforced masonry triplets; (**a**) non-reinforced masonry triplets, (**b**) Retrofitted with diagonal CFRP strips (X form pattern); (c) Retrofitted with CFRP strips parallel to mortar.

**Table 1 polymers-14-03707-t001:** Some mechanical properties of bricks.

Unit Weight (kg/m^3^)	Compressive Strength (MPa)	Young’s Modulus (MPa)	Poisson’s Ratio
1371	24.20	10,000.13	0.2

**Table 2 polymers-14-03707-t002:** Flexural and compressive strength tests for mortars.

Type of Mortar	Compressive Strength [MPa]	COV [%]	Flexural Strength [MPa]	Young’s Modulus E (MPa)
Mortar A (1:1:3)	7.187	5	3.341	3639.24
Mortar B (1:1:5)	3.643		1.453	1821.87

**Table 3 polymers-14-03707-t003:** Mechanical properties of the reinforcing system CFRP strips.

Properties	Value
Width CFRP (mm)	15
Thicknes (mm)	2.5
E_CFRP_ (MPa)	165,000
F_tCFRP_ (MPa)	3100
Rupture strain (%)	1.7

**Table 4 polymers-14-03707-t004:** Shear strength parameters from triplet shear test.

Specimens	Mortar Type	$\sigma_{n}\;\left(MPa\right)$	${\sigma^{\prime }}_{cm}\;\left(MPa\right)$	${f^{\prime }}_{cb}\;\left(MPa\right)$	$F_{max}\;\left(kN\right)$	$\tau_{u}$	*C*	*φ*
ST00A	A	0	7.19	25	16.41	0.71	0.762	0.57
ST02A		0.2			26.11	1.13		
ST06A		0.6			31.67	1.37		
ST10A		1			37.9	1.64		
ST00B	B	0	3.64	25	10.17	0.44	0.435	1.04
ST02B		0.2			20.8	0.9		
ST06B		0.6			24.49	1.06		
ST10B		1			37.42	1.62		

Note: $\;{\sigma^{\prime }}_{cm}\;,{f^{\prime }}_{cb}$ uniaxial compressive strength of mortar and brick, respectively; $\tau_{u}$: ultimate shear strength;$\;\sigma_{n}$: normal pre-compression;$\;F_{max}\;:\mathrm{failure}\;\mathrm{load}\;$; *C*: interface cohesion; *φ*: dilation angle.

**Table 5 polymers-14-03707-t005:** Ultimate shear strength of non-strengthened and strengthened masonry shear triplet test.

Type of Mortar	Specimens Type	Ultimate Shear Strength (MPa)	Improvement Percentage (%)	Ultimate Displacement	Improvement Percentage (%)
A	STC (113T)	0.71	-	2.43	-
	SRV (113RV)	0.76	7	3.71	52.67
	SRX (113RX)	0.96	35.21	3.93	61.72
	SR2X (113R2X)	1.47	107	4.63	90.53
B	STC (115T)	0.44	-	2.64	-
	SRV (115RV)	0.5	14	3.37	27.65
	SRX (115RX)	0.6	36	2.79	5.68
	SR2X (115R2X)	1.1	150	3.9	47.72

**Table 6 polymers-14-03707-t006:** Mechanical properties of the masonry unit and mortar.

Elastic Parameters			Plasticity Parameters	
	Brick	Mortar(B)		
Density (γ) (kg/m^3^)	2200	1800	dilatancy angle (ψ)	20
Young Modulus (E) (MPa)	10,000	1880	Eccentricity parameter (e)	0.1
Poisson’s ratio (*μ*)	0.2	0.18	$\mathrm{Bi}\;\mathrm{and}\;\mathrm{unidirectional}\;\mathrm{compression}\;\mathrm{resistance}\;\mathrm{ratio}\;(\frac{f_{bo}}{f_{co}})$	1.16
			Stress ratio in the meridian in tension (k)	0.67
			Viscosity parameter (ν) (m^2^/s)	0.001

**Table 7 polymers-14-03707-t007:** Mechanical properties of brick–mortar interface (contact cohesive behavior).

Parameters		Magnitude
Normal stiffness (Knn)		40
Shear stifness (Kss)		16
Friction Coefficient		1.04
Damage Initiation (N/mm^2^)	Normal	2
	Shear I	0.44
	Shear II	0.44
Evolution Fracture energies (Nmm/mm^2^)	G_f_^c^	-
	G_F_^I^	0.018
	G_F_^II^	0.2

Knn and Kss, represent the stiffness coeffecients in normal and two shear directions, MN/m. G_f_^I^ is the mode I fracture energy, G_f_^II^ is the mode II fracture energy, and G_f_^c^ is the compressive fracture energy.

**Table 8 polymers-14-03707-t008:** Comparison between numerical results and experiment data for unreinforced shear triplet prisms.

Horizontal Stress σn (MPa)	Shear Bond Strength *τ* (MPa)		Percentage of Error (%)	Ultimate Displacement (mm)		Percentage of Error (%)
	Experiment Results	Numerical Results		Experiment Results	Numerical Results	
0 N/mm^2^	0.71	0.68	3	2.87	2.8	7
0.2 N/mm^2^	0.9	0.86	4	3.1	3.06	4
0.6N/mm^2^	1.06	0.97	9	4.54	4.55	1
1 N/mm^2^	1.62	1.88	26	4.09	4.58	49

**Table 9 polymers-14-03707-t009:** Comparison between numerical results and experiment data for reinforced shear triplet prisms.

Type of Retrofitting	Shear Bond Strength *τ* (MPa)		Percentage of Error (%)	Ultimate Displacement (mm)		Percentage of Error (%)
	Experiment Results	Numerical Results		Experiment Results	Numerical Results	
Vertical strips double sides	0.76	0.76	0	4.71	4.52	19
Diagonal strips(X) double side	1.47	1.68	21	5.1	4.96	0.5

## Data Availability

Not applicable.

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
