# Peer review of "Experimental Research and Numerical Analysis of CFRP Retrofitted Masonry Triplets under Shear Loading"

_polymers, 2022, doi:10.3390/polym14183707_

Round 1

Reviewer 1 Report

Through experimental research and numerical simulation analysis, this paper studies the shear performance of CFRP reinforced masonry structures. The current work has been well designed, and the rich experimental and simulation research results also contain some key information. However, there are still some problems in the writing, which need to be further improved. Please see the following specific comments. 

1# Abstract, please provide some qualitative and quantitative analysis and discussion of strength, post-peak behavior, as well as changing failure modes and ductility from FEM. In addition, please indicate the model accuracy and consistency between FEM and experiments.

2# Introduction, without the basis analysis on the performance, types, advantages and application of FRP, it is somewhat abrupt to directly mention principal role and application of the FRP system in strengthening masonry. The introduction of FRP basic information can provide readers with the understanding of its importance in structural engineering reinforcement. Please review the latest research on some performance, types, advantages and applications of FRP below. Composite Structures, 2020; 246: 112418. Thin-Walled Structures, 2021, 158: 107176. Materials and Structures, 2020, 53: 73.

3# Introduction, for the FRP strengthened masonry or concrete structures, the reinforcement method plays an important role in the stress transfer and failure mode. The main reinforcement methods include ordinary external reinforcement and prestressed reinforcement. Therefore, the relevant summary on the above two reinforcement methods are also crucial for the current research work.

4# The title of Part 2 is a little confused. The title should be materials and methods, of which part 2.1 is materials and part 2.2 is test methods (it is shear triplet tests setup and procedure).

5# Please improve the quality of picture 2 and replace it with a high-definition picture. In addition, from Figure 3, the reviewer did not see the CFRP strip. It is suggested to provide a representative picture.

6# The title of Part 3 should be “results and discussion”.

7# The clarity of Figures 4 to 12 is not enough. Please replace them with high-definition pictures. In addition, in the figure, please give some signs and hints of key areas, such as crack cracking, debonding, etc.

8# In Figure 5, please provide the explanation on curve variations in the decline and rise stages.

9# What are the effects of two different mortar types on the ultimate failure load in Table 4? Please provide relevant analysis and explanation from two aspects: material and interface bonding performance.

10# It can be found from figure 7 that the main failure mode between CFRP and masonry structure is interface debonding. Is this because the performance of the mortar used at the interface is relatively poor? Why not use epoxy resin?

11# Authors should carefully check some small problems of writing norms, such as is should be “Improvement percentage (%)”, capitalization of word initials in Table 5, KN should be “kN” in Table 4, and so on.

12# In figure 10 and 11, FEM are compared with the experimental results. In addition to the comparison of failure mode and bearing performance, the stress distribution is also very critical. It is suggested the authors consider using strain gauge to monitor the stress distribution during the failure process in future research, so as to further compare with the finite element simulation results.

13# The conclusion should be further rewritten, including only 3-4 key information points. The present writing is not the conclusion but the result.

14# From the main research work of this paper, it can be seen that the title is not appropriate, because it does not contain all contents. It should be experimental research and numerical Analysis of CFRP……

Author Response

The authors would like to thank the Editor for the efforts and time in reviewing this paper. 

Reviewer 2 Report

1- The introduction section is weak, and the authors should use more articles to introduce methods.

2- The authors should show a process in the form of a flowchart at the end of the introduction.

3- In figure two: 

The authors explain the connection of the elements to the ground and each other and how to model them.

4- In figures 4 and 7:

 The authors explain the subgroups in the photo and must provide the explanations in text form.

5- In figures 5, 8, and 12:

The borders around the figures should be removed, and the thickness of the curves should be reduced.

6- In Figure 6:

The authors have used a one-variable linear equation and determined the R2 coefficient. Authors should use the multiple linear regression equation. (because it is influential in determining the shear force of many parameters)

7- The quality of all forms should be improved.

8- Figures 10 and 11 should be presented in order because the reader will be confused when reading them. There is a need for a better presentation, and its arrangement should also be improved.

9- The main results should be placed in the conclusion section, and the partial results should be in the results section.

10- After determining the amount of shear, the authors should provide a solution to increase the shear strength in the structure so that this article can be used practically in the construction industry.

11- The authors should compare the results with other articles and research so that the accuracy of the results can be evaluated.

12- A sensitivity analysis is suggested for the effective parameters on the shear force, and the authors consider the impact of each parameter.

13- To determine the level of accuracy in Figure 12, the authors should use the following statistical indicators.

MSE, RMSE, AAE

14- Authors should state the advantages of CFRP over GFRP in this paper.

Author Response

(The authors gave the same response as above.)

Round 2

Reviewer 1 Report

..

Reviewer 2 Report

The Authors responded to all comments.